# Stiffness Evaluation of Laboratory and Plant Produced Foamed Bitumen Warm Asphalt Mixtures with Fiber Reinforcement and Bio-Flux Additive

**DOI:** 10.3390/ma16051950

**Published:** 2023-02-27

**Authors:** Marek Iwański, Anna Chomicz-Kowalska, Krzysztof Maciejewski, Karolina Janus, Piotr Radziszewski, Adam Liphardt, Maciej Michalec, Karol Góral

**Affiliations:** 1Department of Transportation Engineering, Faculty of Civil Engineering and Architecture, Kielce University of Technology, Al. Tysiąclecia Państwa Polskiego 7, 25-314 Kielce, Poland; 2Institute of Road and Bridges, Faculty of Civil Engineering, Warsaw University of Technology, Al. Armii Ludowej 16, 00-637 Warsaw, Poland; 3Zakład Robót Drogowych DUKT Sp. z o.o., 26-052 Nowiny, Poland

**Keywords:** WMA, foamed bitumen, Bio-Flux, complex stiffness modulus, plant production, highly modified bitumen, polymer modified bitumen, high stiffness modulus asphalt concrete, surface course

## Abstract

The present paper investigates the viscoelastic stress-strain responses of laboratory and plant produced warm mix asphalt mixtures containing basalt fiber dispersed reinforcement. The investigated processes and mixture components were evaluated for their efficacy in producing highly performing asphalt mixtures with decreased mixing and compaction temperatures. Surface course asphalt concrete (AC-S 11 mm) and high modulus asphalt concrete (HMAC 22 mm) conventionally and using a warm mix asphalt technique with foamed bitumen and a bio-derived fluxing additive. The warm mixtures included lowered production temperature (by 10 °C) and lowered compaction temperatures (by 15 °C and 30 °C). The complex stiffness moduli of the mixtures were assessed under cyclic loading tests at combinations of four temperatures and five loading frequencies. It was found that the warm produced mixtures were characterized by lower dynamic moduli than the reference mixtures in the whole spectrum of loading conditions, however, the mixtures compacted at the 30 °C lower temperature performed better than the mixtures compacted at 15 °C lower temperature, specifically when highest testing temperatures are considered. The differences in the performance of plant and laboratory produced mixtures were ascertained to be nonsignificant. It was concluded that the differences in stiffness of hot mix and warm mixtures can be attributed to the inherent properties of foamed bitumen mixtures and that these differences should shrink in time.

## 1. Introduction

The development of road infrastructure can stimulate economic growth on one hand [1,2], but on the other hand the ongoing economic growth and globalization of the economies also universally increase the traffic loads exerted on the existing road networks [3,4,5]. Hence, new methods and processes are currently under development for producing sustainable, long-lasting pavements that would have a more manageable impact on the environment.

The present study investigates warm mix asphalts (WMA) intended for producing surface, binding, and road base layers in high performing and long-lasting pavements. The investigated production technique involves the utilization of decreased production and paving temperatures enabled by a bio-derived fluxing additive and foaming of asphalt binder. The high performance of the mixtures is provided by using polymer modified asphalt binders and dispersed fiber reinforcement. 

Decreasing the processing temperatures of asphalt mixtures by exploiting the water foaming phenomenon may be executed either by asphalt binder foaming using zeolites and other similar materials capable of encapsulating water and its controlled release [6,7,8,9] or direct water injection into the asphalt binder (also referred to as “mechanical foaming”) [10,11]. Asphalt binder foaming by injection of water has been utilized with great success for producing warm asphalt mixtures [12,13,14,15,16], with its effects typically being owed to the increased volume of the binder, which enhances the mixture coatability and the shear-thinning mechanism caused by reduced binder film thickness in the foam [17]. In previous studies it has been shown that the impact of water foaming on the performance properties and aging characteristics of asphalt binders is small in technical terms and statistically non-significant in many of the investigated cases. In particular, no evidence for significant oxidative stress associated with asphalt binder foaming was found [18], and while the effects of foaming on viscoelastic properties of asphalt binders may be significant [19], they subside after short-term aging [20].Therefore, the effects of foaming on asphalt binders may be omitted in the mixture design, hence simplifying it and not interfering with other possible additives. 

Other commonly investigated techniques for producing WMA mixtures is the use of fluxing agents. The high adequacy of using the bio-derived fluxing additive (Bio-Flux) for reducing processing temperatures of asphalt mixtures with polymer modified bitumen was shown in [21]. The additive lowers the viscosity of the asphalt binder blend, allowing for the decrease in processing temperatures at production, but with time it causes an increase in stiffness and the performance of the asphalt mixture [22,23]. Other notable plant derived fluxing additives and their uses include Oleoflux and Green Seal [24], waste cooking oils [25,26,27] and tall oils [28].

Warm mix asphalts, regardless of the production technique are known to differ in mechanical properties from their HMA counterparts. In field studies [16] it was found in this respect that WMA mixtures are typically characterized by lower dynamic stiffness moduli and the asphalt binders in these mixtures are less affected by aging due to lower processing temperatures [29,30]. However, the magnitude of these effects was observed to decrease in time and no specific distress was recorded in WMA test sections. These findings were further confirmed by laboratory [31,32,33] and other field studies [34,35].The efficacy of fiber reinforcement in asphalt mixtures has been shown in several studies to date. Basalt fiber reinforcement increases the stiffness of asphalt mixtures at high service temperatures without impacting their performance at low temperatures and improves their fatigue life [36,37]. In turn, the use of basalt fibers has a positive impact on the service life of asphalt pavements and needs for their rehabilitation [38]. Similar effects were found when using aramid [39] and carbon and glass fibers [40,41]. In warm asphalt mixtures the addition of fiber reinforcement also improves their stress-strain characteristics as it was shown in terms of resilient modulus [42], permanent deformations [42,43] and fatigue life [42,44].

Based on the presented state of the art, the described above technique was investigated for its capability for producing and paving high performing asphalt mixtures at considerably lowered temperatures.

## 2. Materials and Methods

### 2.1. Asphalt Mixtures

The two types of asphalt concrete mixtures evaluated in this study included surface course asphalt concrete (AC-S) and high modulus asphalt concrete (HMAC), produced as typical HMAs and as WMAs with foamed bitumen and a fluxing additive. The basic characterization of the investigated mixtures is shown in Table 1.

Details regarding the composition of the investigated mixtures are presented in Table 2. The asphalt mixtures were based on crushed limestone and gabbro aggregates, the latter being utilized in the coarse fractions of the surface course mix. The grain size distribution in both types of investigated mixtures is presented in Figure 1. The mixtures’ grading conformed to the requirements of domestic requirements [45] represented in the figure by the respective grading limits. 

The asphalt binder content amounted to 5.4% (m/m) in the AC-S mixtures and 5.0% in HMAC formulations. An adhesion promoter was added to the bitumen prior to mixing with the aggregates or prior to asphalt binder foaming at a 0.3% rate per bitumen mass. The study included the use of cut basalt fibers, 12 mm (AC-S mixes) and 24 mm (HMAC mixes) in length and approx. 0.03 mm in diameter, added at 0.3% and 0.2% rates per final asphalt mix, respectively. The compositions of the investigated mixtures (grading, asphalt binder content, fiber content, fiber length) were optimized in the course of the preliminary work for this study.

The details regarding the processing of the investigated asphalt mixtures are provided in Table 3. The asphalt mixtures produced as HMAs utilized asphalt binder that was added conventionally to the mineral mixture, whereas the warm mix asphalts included foamed asphalt binders produced either as in [46] when mixed in laboratory, or produced in an industrial scale asphalt plant with a Green Pac (Astec Inc., Chattanooga, TN, USA) foaming system as shown in Figure 2a. Figure 2b shows the weighed bags with the basalt fibers to be manually added to the mineral mixture in the asphalt mixer at the asphalt plant.

Mixtures produced in laboratory were subjected to short-term oven aging (STOA) at 135 °C for 2 h with mixing after 1 h and 2 h [45]. Mixtures produced in the asphalt plant were collected upon transfer to the paver. The mixing and compaction temperatures of HMAs were set according to the guidelines of the asphalt binders’ manufacturer, while the production of WMAs was conducted at a 20 °C lower temperature. Compaction of WMA mixtures was done in temperatures 15 °C and 30 °C lower than the HMAs.

### 2.2. Asphalt Binders

The investigated surface course and high modulus asphalt mixtures were produced with a 45/80-80 highly modified asphalt binder and a 25/55-60 polymer modified asphalt binder respectively. The asphalt binders used in warm mix formulations were modified by the addition of a bio-derived Bio-Flux additive. The additive is produced by subjecting fatty acid methyl esters to oxidation in the presence of cobalt catalyst and cumene hydrogen peroxide. Details on the properties and derivation of this additive, as well as on preparing the modified asphalt binders are described in [46]. The FTIR evaluation of the additive and asphalt binders is provided in [47]. The properties of the asphalt binders used in the study are shown in Table 4. The contents of the fluxing additive were optimized in preliminary the work for this study.

### 2.3. Complex Stiffness Modulus Testing

The characterization of viscoelastic properties of the asphalt mixtures was conducted based on the results of cyclic uniaxial compressive tests in accordance with the methodology provided by AASHTO T 342 standard. The experiments used cylindrical samples 100 mm in diameter and 150 mm in height, cored and sawed from 150 × 200 mm gyratory compactor specimens, compacted to a target of 3% air void content (2–4% specification range). The testing was conducted at four temperatures (−10 °C, 5 °C, 20 °C and 35 °C) and six loading frequencies (0,1, 0,5, 1, 5, 10, 25 Hz). The experiments included 4 replicates (samples) per mixture, each with 3 LVDT’s. The tests were conducted in controlled stress mode, with stresses set to induce a strain level in the range of 50–100 µε to ensure linear viscoelastic response [48].

Complex stiffness modulus master curves for each of the tested mixtures were constructed by shifting the data measured in respective temperatures to new (*f_fict_*, fictional) frequencies at a set reference temperature (*T_ref_*). The principle of this operation is shown in Figure 3. Shifting of the obtained data was conducted by assuming the time-temperature superposition principle [49,50], which provides an equivalence of the temperature and loading time effects in the form of a shift factor.

The values of complex stiffness moduli obtained at different temperatures were shifted in the frequency domain using the Williams-Landel-Ferry shift factor (1), dependent on a testing temperature (*T*), reference temperature (*T_ref_ =* 10 °C), and two empirical constants (*C_1_, C_2_*) to produce continuous master curves describing a relationship between the dynamic modulus of the mixtures and a new shifted (fictional) frequency [51,52,53,54]. The relationship between the actual and fictional frequencies (2) was derived from [54]. The measured data were fitted to a sigmoidal mathematical model (3) by the method of least squares optimization [55,56].
(1)$$log\;a\left(T\right)=-\frac{C_{1}\left(T-T_{ref}\right)}{C_{2}+T-T_{ref}}$$
(2)$$log\;f_{fict}-log\;f=log\;a_{T}$$
(3)$$log\;|E^{*}|=\delta +\frac{\alpha }{1+\exp\left(\beta +\gamma \cdot logf_{fict}\right)}$$

## 3. Results

### 3.1. Laboratory Produced Mixtures

The complex stiffness modulus master curves produced by frequency-shifting the experimental data and fitting to the sigmoidal models for the investigated laboratory produced mixtures are shown in Figure 4. The plot showing measured versus predicted values of the complex stiffness moduli is presented in Figure 5. The shifted data spanned approx. from 2.0 × 10^−5^ to 3.7 × 10^5^ Hz and from 9.0 × 10^−5^ to 4.1 × 10^5^ Hz in case of the AC-S and HMAC mixtures respectively. The presented curves show that the investigated factors, i.e., fiber reinforcement and production technique had similar effects on both types of the mixtures—AC-S and HMAC. The addition of fiber reinforcement resulted in an increase in the complex stiffness moduli in the entire range of the testing conditions, positioning the master curves of *H Fib* visibly above others, the reference mixtures in particular. The change of the production technique of the fiber containing mixture to WMA, by decreasing the production temperature, introducing foamed bitumen into the mix, and lowering the compaction temperature caused a significant drop in the complex stiffness moduli in the entire range of frequencies. It should be noted however, that changing the compaction temperature from 130 °C to 115 °C had some effect, but which by the inspection of the master curves was only hardly visible. Another notable observation could be linked to the AC-S mixtures and the effect of fiber reinforcement—the fiber containing mixtures exhibited significantly higher values of complex stiffness moduli at the low temperature-high frequency end of the investigated spectrum than the reference mixture.

The diagnostic plot in Figure 5 shows that the complex stiffness moduli values are evenly distributed around the identity line, although particularly in the case of the HMAC mixtures the increase in the stiffness of the materials resulted in increased spread of the measured stiffness values. The high R^2^ values (>0.98) indicate good overall fit of the data to the sigmoidal models.

It is well established that volumetric relationships in asphalt mixtures may significantly impact their stress-strain responses [57,58]. In regards to the present study, the effects of air void content on the dynamic modulus of the tested mixtures were a concern, particularly the possible decrease of dynamic modulus with increased air void contents [59]. Therefore an analysis was conducted to evaluate the potential significance of these effects on the obtained results. Figure 6 shows air void contents in the tested samples versus the measured values of complex stiffness moduli. The air void contents of all tested samples were contained in the 2.3–3.7% range, with V_a_ means differing by no more than 0.9%. The analysis of variance did not confirm the significance of the V_a_ related effect on the dynamic modulus (p-value > 0.05). At the same time the effects of mixture type were found to be statistically significant.

The values of the complex stiffness moduli of the evaluated mixtures measured at 10 Hz shown in Figure 7 confirm the observations made by inspecting the master curves in Figure 4. In case of all temperatures and in both types of mixtures the highest values of complex stiffness modulus were recorded by the *H Fib* formulations. Regarding the WMAs, in most cases the mixtures compacted at temperatures 30 °C lower than the reference ones were characterized by the lowest values of complex stiffness modulus. It should be noted however, that usually the confidence intervals for the means of the *W Fib -15* and *W Fib-30* overlapped significantly. To evaluate the differences between the performance of specific mixtures, post-hoc multiple comparison tests were conducted (Table 5). Based on the results of these tests it can be concluded that in most cases the differences between the *H Ref* and *H Fib* mixtures and the *W Fib-15* and *W Fib-30* were not statistically significant as shown in the first and last row of the table. On the other hand, the warm mix *Fib* mixtures were significantly different in terms of their dynamic moduli from the hot mix fiber containing mixtures. It should also be recognized that the *AC-S W Fib-30* and *HMAC W Fib-30* mixes performed similarly to the reference mixtures at two (−10 °C, 20 °C) and one (−10 °C) temperatures respectively.

Based on these findings, the reference mixes, *AC-S W Fib-30* and *HMAC W Fib-30* mixtures were selected to be produced at the asphalt plant and paved.

### 3.2. Plant Produced Mixtures

The asphalt concrete mixtures produced at the asphalt plant were subjected to the same testing as laboratory produced mixtures. Figure 8 presents the complex stiffness modulus master curves derived for the plant produced mixtures and their laboratory produced counterparts (as shown before) for direct comparisons. The measured versus predicted dynamic moduli of the plant produced mixtures are shown in Figure 9.

The magnitude of differences between the viscoelastic responses of plant and laboratory produced material changed depending on the loading times and testing temperature. At the high end of the frequency spectrum the plant produced surface course mixtures (P-AC-S) were characterized by higher values of complex stiffness modulus, but the differences diminished at low frequencies. In the case of the high modulus mixtures, the plant produced mixtures (P-HMAC) had lower dynamic moduli than the laboratory produced mixtures at the high end of the frequency spectrum, then in the 0.1 to 10 Hz range these values were similar and eventually at the lowest frequencies the plant mixtures appeared stiffer than the laboratory produced ones.

The differences between the plant produced reference and warm mix asphalts were consistent throughout the frequency range with the WMAs being characterized by lower values of complex stiffness moduli.

The diagnostic plot in Figure 8 indicates good overall fit of the data to the sigmoidal models (R^2^ > 0.99) and shows that the complex stiffness moduli values were evenly distributed around the identity line.

The relationships between the air void content of the samples from plant produced mixtures and their complex stiffness moduli are investigated in Figure 10. The air void contents of all tested samples ranged from 2.5% to 3.2%, and the largest difference between the means in the respective mixtures amounted to 0.4%, which considering the differences in the dynamic moduli can be viewed as minute. The analysis of variance in fact did not confirm the effect of air void content to be a significant factor affecting the dynamic moduli of the plant produced mixtures (*p*-value > 0.05).

Figure 11 shows that at the frequency of 10 Hz, the measured values of dynamic moduli in the plant-produced mixtures and their laboratory produced counterparts (as shown in Figure 7, for comparisons) reflect the relationships seen in the master curves provided in Figure 8. In the surface course mixtures, the plant produced material was on average stiffer than the laboratory produced mixtures, whereas in the HMAC mixtures this was the case only at 20 °C and 35 °C.

To assess the significance of the presented differences, a post-hoc statistical analysis was again employed. The most significant observations in this respect can be made by inspecting the first two rows of Table 6, evaluating the differences between the laboratory and plant-produced mixtures. As shown in the second row, the performance of plant and laboratory produced WMAs was mostly similar, with statistically significant differences observed only at one of the four temperatures for each mix (5 °C for the AC-S mix and 35 °C for the HMAC mix). The reference HMAC mixtures also performed comparably, while the performance of laboratory and plant produced surface course mixtures was significantly different in three out of four temperatures (first row).

## 4. Discussion

The investigated mixtures were controlled in terms of their mineral composition and asphalt binder content. The air void content was also controlled during the compaction process of the mixtures and the statistical analysis concluded that variability in V_a_ had negligible effects on the measured viscoelastic responses. It is well established that the stiffness of typical HMA asphalt mixtures depends primarily on the properties of asphalt binder, properties of the mineral aggregates and the volumetric relationships between the constituents of the asphalt mixture [55,57,59]. Among those factors, the asphalt binder, specifically the effects of the fluxing agent, remains suspect to be a cause of the inferior performance of the tested WMA mixtures. It is, however, difficult to unquestionably and directly link the effects of the fluxing additive to the decreased stiffness of the asphalt mixtures. Although the addition of the fluxing agent had some detrimental effects on the performance characteristics of the 25/55-60 asphalt binder used in the HMAC mixtures, its effects in the 45/80-80 binder were negligible and could even be considered as positive. At the same time both types of mixtures, containing either of the binders were negatively and to a similar extent affected by the WMA processes.

With the aforementioned factors plausibly excluded and controlled for, another group of effects, linked strictly to the processing of the asphalt mixtures can be named. The effects of action of high temperature during production (leading to asphalt mixture stiffening due to asphalt binder aging and binder absorption), production temperature and the use of foamed bitumen as a binder differentiated the affected mixtures.

Field experiences [16] have shown that the utilization of warm mix techniques, due to the lowered processing temperatures, may significantly affect the high temperature performance of the asphalt binder contained in the WMAs. This has been shown to have a negative effect on the early service performance of WMAs [31] and is expected due to the aforementioned relationships between the stiffness of asphalt binders and asphalt mixtures. Additionally, a different study [60] has shown that warm asphalt mixtures, specifically those produced with water-injected foamed bitumen, yield lower (by 0.27 percent point on average) optimum design binder content than comparable HMA mixtures. This fact is attributed by Williams and Braham [60] to lower reported binder absorption in foamed WMAs in their early service life. This being the case, it leaves more free asphalt binder in the mixture at the time of paving and compaction, but may further hinder short-term high temperature performance. The observations of lower asphalt binder content applied in practice of foamed WMA mixtures was confirmed in different WMA projects [14], which however was also linked to increased cracking of such layers and should be avoided.

## 5. Conclusions

The present study investigated surface course asphalt concrete (AC-S) and high modulus asphalt concrete (HMAC) mixtures produced in three variants: As HMA reference mixtures, fiber reinforced HMAs, and fiber reinforced WMAs produced with foamed bitumen and fluxing agent. The first part of the study concerned asphalt mixtures produced in the laboratory, while plant-produced asphalt mixtures were investigated in the second. The presented results can be summarized as follows:− fiber reinforcement in HMAs resulted in an increase in the dynamic moduli of both AC-S and HMAC mixtures,− the fiber reinforced WMA mixtures exhibited lower dynamic moduli than the HMA mixtures, in both laboratory and plant produced variants,− the compaction temperature of the WMA mixtures was not a statistically significant effect in terms of their complex stiffness modulus |E*|,− the respective plant and laboratory mixtures performed similarly in most cases – except for the AC-S mix.

While the observed differences in the dynamic moduli of the HMA and WMA mixtures could not be easily attributed to the effects of bitumen foaming or the presence of the Bio-Flux additive in the asphalt binders, they were similar to those reported by other authors investigating different types of warm asphalt mixtures.

Future studies in the area should investigate the performance of fiber reinforced asphalt mixtures subjected to long-term aging, specifically in the scope of the effects of binder absorption and binder aging.

## Figures and Tables

**Figure 1 materials-16-01950-f001:**
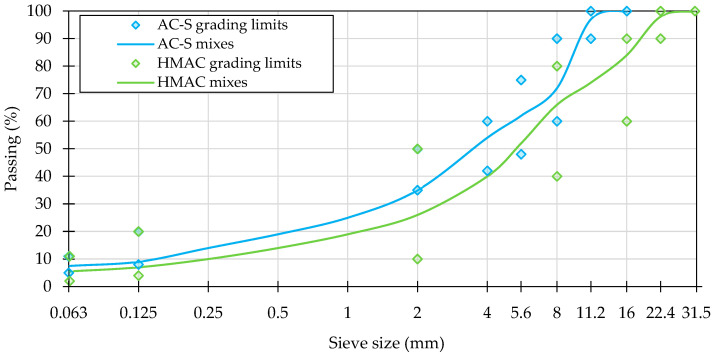
Grain size distribution in the investigated asphalt mixtures.

**Figure 2 materials-16-01950-f002:**
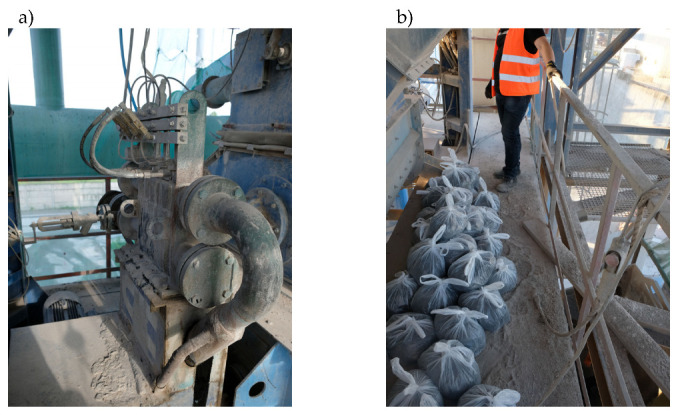
Plant production of asphalt mixtures: Green Pac asphalt foaming system (**a**), batches of basalt fibers prepared for mixing (**b**).

**Figure 3 materials-16-01950-f003:**
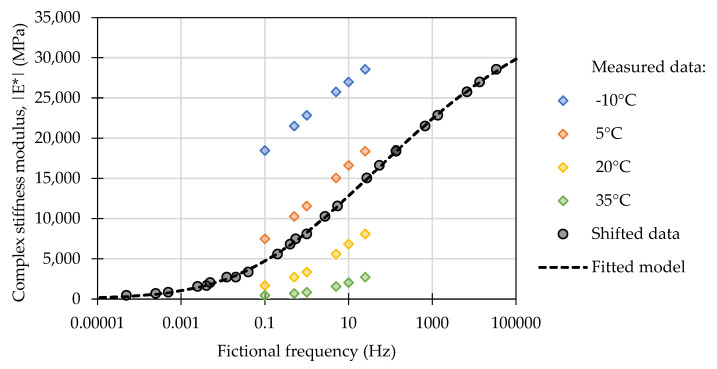
The principle of shifting the measured values of the complex stiffness moduli in the frequency domain to construct a master curve.

**Figure 4 materials-16-01950-f004:**
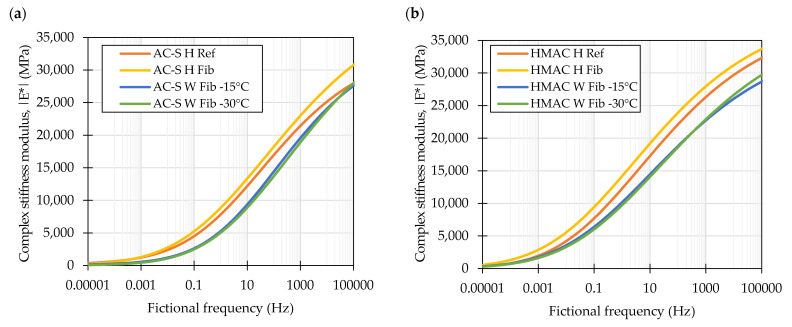
Complex stiffness modulus master curves characterizing the evaluated mixtures produced in laboratory: Surface course mixtures (**a**) and high modulus mixtures (**b**).

**Figure 5 materials-16-01950-f005:**
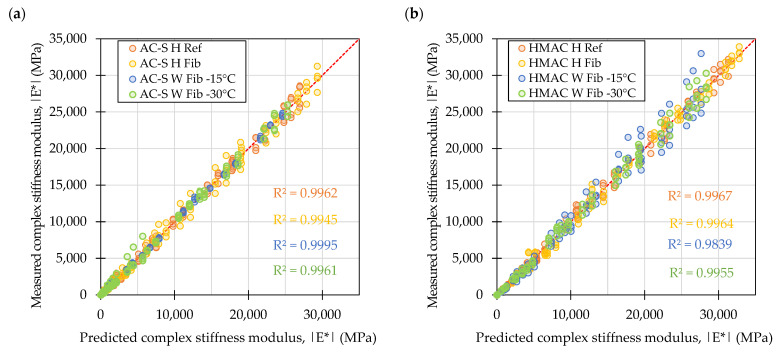
Measured versus predicted values of complex stiffness moduli in samples from laboratory produced mixtures: Surface course mixtures (**a**), high modulus asphalt concrete mixtures (**b**).

**Figure 6 materials-16-01950-f006:**
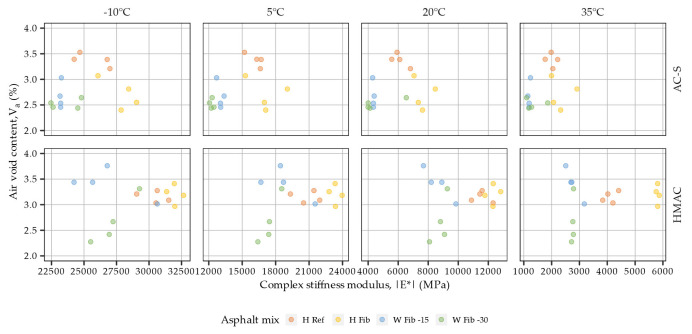
Complex stiffness moduli measured at 10 Hz versus air void content in laboratory produced samples from surface course and high modulus asphalt concrete.

**Figure 7 materials-16-01950-f007:**
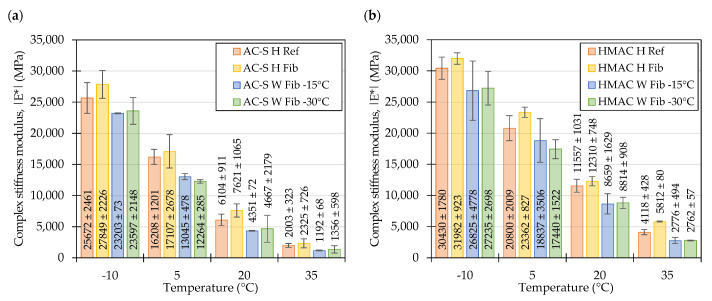
Complex stiffness moduli of laboratory produced mixtures measured at 10 Hz: Surface course mixtures (**a**), high modulus asphalt concrete mixtures (**b**), means and 95% confidence intervals (also shown as error bars).

**Figure 8 materials-16-01950-f008:**
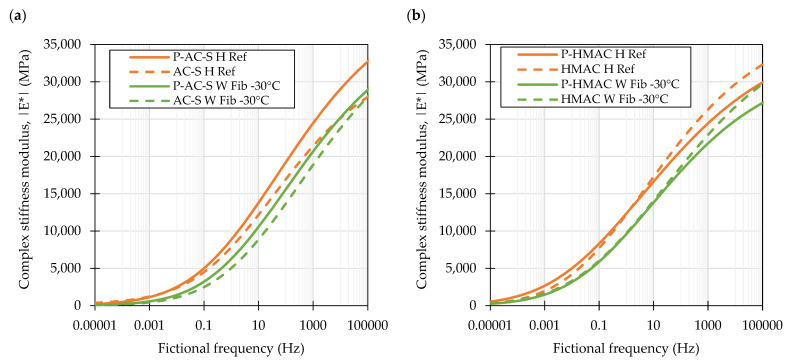
Complex stiffness modulus master curves characterizing the evaluated surface course (**a**) and high modulus asphalt concrete (**b**) mixtures produced in the asphalt plant.

**Figure 9 materials-16-01950-f009:**
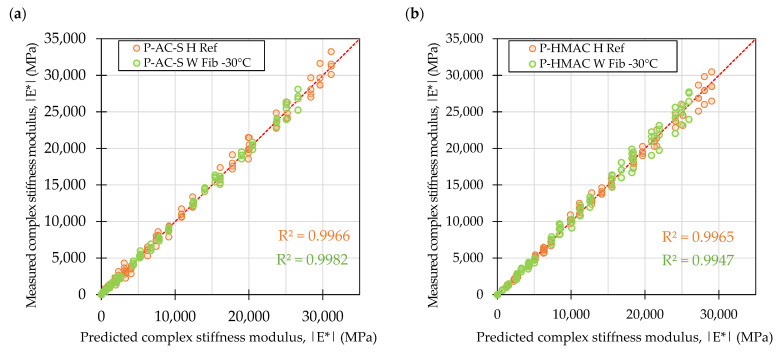
Measured versus predicted values of complex stiffness moduli in samples from plant produced mixtures: Surface course mixtures (**a**), high modulus asphalt concrete mixtures (**b**).

**Figure 10 materials-16-01950-f010:**
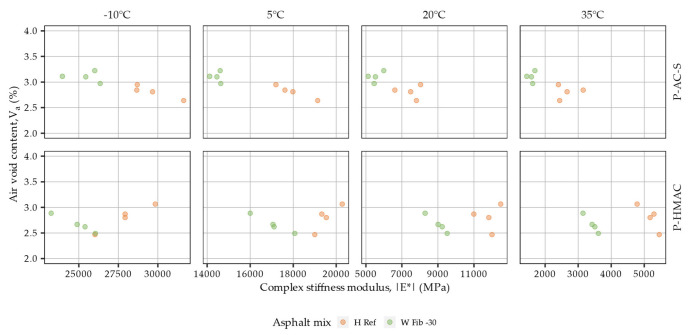
Complex stiffness moduli measured at 10 Hz versus air void content in plant produced samples from surface course and high modulus asphalt concrete.

**Figure 11 materials-16-01950-f011:**
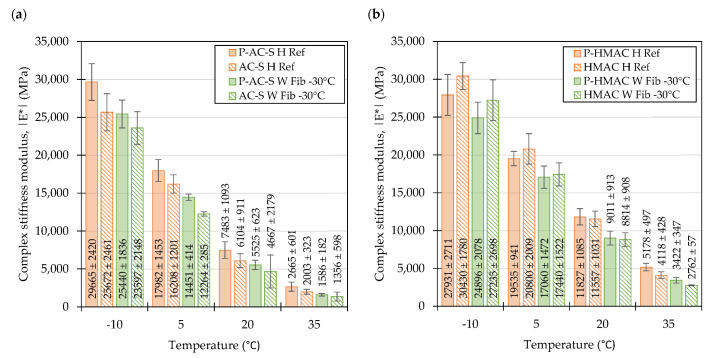
Complex stiffness moduli of plant produced surface course (**a**) and high modulus asphalt concrete mixtures (**b**) measured at 10 Hz; means and 95% confidence intervals (also shown as error bars).

**Table 1 materials-16-01950-t001:** Specifications of the investigated mixtures.

Mixture Designation	Mix Type	Fibers Added	Production Technique	Compaction Temperature (°C)	Mix Production
AC-S H Ref	AC-S	No	HMA	145	Laboratory
AC-S H Fib	AC-S	Yes	HMA	145	Laboratory
AC-S W Fib-15	AC-S	Yes	WMA	130 (−15)	Laboratory
AC-S W Fib-30	AC-S	Yes	WMA	115 (−30)	Laboratory
HMAC H Ref	HMAC	No	HMA	145	Laboratory
HMAC H Fib	HMAC	Yes	HMA	145	Laboratory
HMAC W Fib-15	HMAC	Yes	WMA	130 (−15)	Laboratory
HMAC W Fib-30	HMAC	Yes	WMA	115 (−30)	Laboratory
P-AC-S H Ref	AC-S	No	HMA	145	Plant (no STOA)
P-AC-S W Fib-30	AC-S	Yes	WMA	115 (−30)	Plant (no STOA)
P-HMAC H Ref	HMAC	No	HMA	145	Plant (no STOA)
P-HMAC W Fib-30	HMAC	Yes	WMA	115 (−30)	Plant (no STOA)

**Table 2 materials-16-01950-t002:** Composition of the investigated asphalt mixtures.

	AC-S Ref Mixes	AC-S Fib Mixes	HMAC Ref Mixes	HMAC Fib Mixes
Nominal maximum aggregate size (mm)	11	11	22	22
Mineral mix density, ρa (Mg/m3)	2.855	2.855	2.705	2.705
Filler, limestone (%)	5.7	5.7	3.8	3.8
Limestone, 0/2 mm (%)	18.0	18.0	-	-
Limestone, 0/4 mm (%)	14.1	14.1	-	-
Gabbro, 2/5 mm (%)	18.0	18.0	-	-
Gabbro, 4/8 mm (%)	12.2	12.2	-	-
Gabbro, 8/11 mm (%)	26.3	26.3	-	-
Limestone, 0/2 mm (%)	-	-	9.5	9.5
Limestone, 0/4 mm (%)	-	-	19.0	19.0
Limestone, 2/8 mm (%)	-	-	29.2	29.2
Limestone, 8/16 mm (%)	-	-	18.1	18.1
Limestone, 16/22 mm (%)	-	-	15.2	15.2
Asphalt binder content (%, m/m)	5.4	5.4	5.0	5.0
Adhesion promoter, Wetfix BE, Minova Ekochem, (% per asphalt binder)	0.3%	0.3%	0.3%	0.3%
Basalt fibers, Basalttech Sp. z o.o.	-	0.3% (12 mm)	-	0.2% (24 mm)

**Table 3 materials-16-01950-t003:** Processing details of the investigated asphalt mixtures.

	AC-S H Mixes (HMA)	AC-S W Mixes (WMA)	HMAC H Mixes (HMA)	HMAC W Mixes (WMA)
Asphalt binder type	45/80-80 (neat)	Foamed 45/80-80 +3% Bio-Flux	25/55-60 neat	Foamed 25/55-60 +2% Bio-Flux
Foaming water content (%)	-	2.0	-	2.0
STOA (2 h) temperature (°C)	135	135	135	135
Mixing temperature (°C)	180	160	180	160
Compaction temperature (°C)	145	130, 115	145	130, 115

**Table 4 materials-16-01950-t004:** Characterization of asphalt binders used for producing the investigated asphalt mixtures.

	Unit	AC-S Mix		HMAC Mix		Testing Method
		45/80-80 Neat	Foamed 45/80-80 +3% Bio-Flux	25/55-60Neat	Foamed 25/55-60 +2% Bio-Flux	
Penetration at 25 °C	0.1 mm	75	124	40	65	EN 1426
Softening point	°C	95.5	81.2	63.4	58.0	EN 1427
Fraass breaking point	°C	−22	−25	−13	−17	EN 12593
Dynamic viscosity at 135 °C	Pa·s	2.81	1.31	1.70	1.21	EN 13702–2
Dynamic viscosity at 135 °C after RTFOT	Pa·s	3.77	2.40	2.51	2.12	EN 13702–2
High critical temperature (G*/sin(δ) = 2.2 kPa)	°C	81.3	83.1	78.9	77.6	EN 14770, EN 12607-1
Low critical temperature (S = 300 Mpa, m = 0.3)	°C	−22.2, −18.3	−25.8, −22.1	−19.2, −16.0	−21.8,−18.9	EN 14771, EN 12607-1 EN 14769
J_nr 3.2 kPa_ at 60 °C	1/kPa	0.026	0.020	0.080	0.180	EN 16659
%R_3.2 kPa_	%	96	96	70	61	EN 16659

**Table 5 materials-16-01950-t005:** Results of post-hoc Tukey HSD tests (*p*-values) for assessment of differences between the dynamic moduli (|E*|) in mixture pairs; *p*-values < 0.05 denote significant differences in pairs.

Compared Pairs		Surface Course AC mix (AC-S)				High Stiffness Modulus AC Mix (HMAC)			
Mix-1	Mix-2	−10	5	20	35	−10	5	20	35
H Ref	H Fib	0.078	0.477	0.058	0.419	0.575	0.059	0.393	<0.001
H Ref	W Fib-15	0.042	0.001	0.027	0.008	0.044	0.179	<0.001	<0.001
H Ref	W Fib-30	0.096	<0.001	0.075	0.034	0.080	0.012	<0.001	<0.001
H Fib	W Fib-15	<0.001	<0.001	<0.001	0.001	0.005	0.001	<0.001	<0.001
H Fib	W Fib-30	0.001	<0.001	0.001	0.002	0.008	0.000	<0.001	<0.001
W Fib-15	W Fib-30	0.960	0.587	0.930	0.848	0.985	0.433	0.986	1.000

**Table 6 materials-16-01950-t006:** Results of post-hoc Tukey HSD tests (*p*-values) for assessment of differences between the dynamic moduli (|E*|) in mixture pairs; *p*-values < 0.05 denote significant differences in pairs.

Compared Pairs		Surface Course AC Mix (AC-S)				High Stiffness Modulus AC Mix (HMAC)			
Mix-1	Mix-2	−10	5	20	35	−10	5	20	35
P-H Ref	H Ref	0.004	0.004	0.104	0.019	0.091	0.230	0.904	0.000
P-W Fib-30	W Fib-30	0.229	0.001	0.425	0.625	0.120	0.927	0.960	0.004
P-H Ref	P-W Fib-30	0.003	0.000	0.017	0.000	0.035	0.009	0.000	0.000
P-H Ref	W Fib-30	0.000	0.000	0.001	0.000	0.884	0.025	0.000	0.000
H Ref	P-W Fib-30	0.994	0.004	0.716	0.173	0.000	0.000	0.000	0.003
H Ref	W Fib-30	0.155	0.000	0.087	0.022	0.026	0.001	0.000	0.000

## Data Availability

Data available at request.

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
