# Peer review of "Stiffness Evaluation of Laboratory and Plant Produced Foamed Bitumen Warm Asphalt Mixtures with Fiber Reinforcement and Bio-Flux Additive"

_materials, 2023, doi:10.3390/ma16051950_

Round 1

Reviewer 1 Report

This manuscript investigates Stiffness master curves of Bitumen Warm Asphalt Mixtures with Fiber Reinforcement ,and Bio-Flux Additive.  

In all text, the number of references are in Persian and should be modified.

1-      The literature review should be expanded and papers about warm mix technology and mentioned additives should be added.

2-      Where is the “Conclusions” sections in this paper?

3-      What is the specifications of Fiber Reinforcement and Bio-Flux Additive?

4-      As an example, add a case of Master curve drawing method, in the paper.

5-      Conclusions have been made in this article very hastily, it is better to express the conclusion separately for each additive for the laboratory and Plant.

Due to these issues, this manuscript is suitable for publication after major revision.

Author Response

Dear Reviewer,

We would like to voice our gratitude for the time and effort spent revising our paper titled “Stiffness Evaluation of Laboratory and Plant Produced Foamed Bitumen Warm Asphalt Mixtures with Fiber Reinforcement and Bio-Flux Additive”. We truly feel that your remarks are a significant contribution to the overall quality of our paper and enabled us to rectify its shortcomings.

After a thorough revision, we present you the corrected version of the manuscript for its assessment. Please find the detailed responses to your comments below.

Best regards,

Authors

Reviewer 1

This manuscript investigates Stiffness master curves of Bitumen Warm Asphalt Mixtures with Fiber Reinforcement ,and Bio-Flux Additive.  

In all text, the number of references are in Persian and should be modified.

Thank you for the remark. The references are numbered according to the current MDPI reference citing style.

  • The literature review should be expanded and papers about warm mix technology and mentioned additives should be added.

Thank you for the accurate remark. We have expanded the literature review on the effects of WMA techniques on the properties of asphalt mixtures in L72-78. The references for the Bio-flux additive were included in the but the name of the additive was not provided in L64-69 – this has been amended.  

2-      Where is the “Conclusions” sections in this paper?

Thank you for the accurate remark. We have added a conclusions section summarizing the findings.

3-      What is the specifications of Fiber Reinforcement and Bio-Flux Additive?

Thank you for the accurate comment. In L140-143 we have added the basic description of the Bio-Flux additive: “The additive is produced by subjecting fatty acid methyl esters to oxidation in the presence of cobalt catalyst and cumene hydrogen peroxide.“ and the following line contains a reference a previously published open access paper describing the composition, properties and  

The information regarding the fiber reinforcement used is provided in L91-95 in section 2.1 Asphalt mixtures and the source (producer) is provided in the last row of Table 2.

4-      As an example, add a case of Master curve drawing method, in the paper.

Thank you for the comment. We have added an appropriate figure (Figure 3) with and explanation in L159-168.

5-      Conclusions have been made in this article very hastily, it is better to express the conclusion separately for each additive for the laboratory and Plant.

Thank you for the accurate remark. We have added a conclusions section summarizing the findings.

Reviewer 2 Report

This study evaluated the stiffness resistance of WAM produced at two different scales with fibers. In my opinion the article deserves publication in Materials journal and meets the standards of scientific publication. The article is well written but still requires some refinements:

1)     The practical scale significance fiber reinforced WAMs, must be added to the abstract in starting lines.

2)     Literature review must be enriched with studies related to performance WAMs with fibers.

3)     L55, this information does not need three citations. Try to minimize the merged citations or explain them specifically.

4)     Table about the quantitative details of mixes must be given.

5)     Y-scale of figure 5 can be enlarged.

6)     No need for standard deviation in data labels, compression stiffness modulus. Error bars are sufficient.

7)     The results are clearly presented and discussed with validation.

Author Response

Dear Reviewer,

We would like to voice our gratitude for the time and effort spent revising our paper titled “Stiffness Evaluation of Laboratory and Plant Produced Foamed Bitumen Warm Asphalt Mixtures with Fiber Reinforcement and Bio-Flux Additive”. We truly feel that your remarks are a significant contribution to the overall quality of our paper and enabled us to rectify its shortcomings.

After a thorough revision, we present you the corrected version of the manuscript for its assessment. Please find the detailed responses to your comments below.

Best regards,

Authors

Reviewer 2

This study evaluated the stiffness resistance of WAM produced at two different scales with fibers. In my opinion the article deserves publication in Materials journal and meets the standards of scientific publication. The article is well written but still requires some refinements:

Thank you for the generous comment.

  • The practical scale significance fiber reinforced WAMs, must be added to the abstract in starting lines.

Thank you for the accurate remark. We have added appropriate statements in the beginning of the abstract (L15-17):

“The present paper investigates the viscoelastic stress-strain responses of laboratory and plant produced warm mix asphalt mixtures containing basalt fiber dispersed reinforcement. The inves-tigated processes and mixture components were evaluated for their efficacy in producing highly performing asphalt mixtures with decreased mixing and compaction temperatures.”

  • Literature review must be enriched with studies related to performance WAMs with fibers.

Thank you for the accurate remark. We have expended the review in L84-87

3)     L55, this information does not need three citations. Try to minimize the merged citations or explain them specifically.

Thank you for the remark. We have elaborated on this topic and expanded L55-61:

“In previous studies it has been shown that the impact of water foaming on the performance properties and ageing characteristics of asphalt binders is small in technical terms and statistically non-significant in many of the investigated cases. In particular, no evidence for significant oxidative stress associated with asphalt binder foaming was found [18], and while the effects of foaming on viscoelastic properties of asphalt binders may be significant [19], they subside after short-term ageing [20]. Therefore, the effects of foaming on asphalt binders may be omitted in the mixture design, therefore simplifying it and not interfering with other possible additives.”

  • Table about the quantitative details of mixes must be given.

Thank you for the comment. We have provided the quantitative data regarding the composition of the mixtures in Table 2, the  grain size distribution in Figure 1, and the details regarding the production, ageing and compaction in Table 3. Moreover, detailed properties of the asphalt binders are provided in Table 4.

5)     Y-scale of figure 5 can be enlarged.

Thank you for the remark. We have increased the y-scale of Figures 6 and 10 (Figures 5 and 9 in the original manuscript).

6)     No need for standard deviation in data labels, compression stiffness modulus. Error bars are sufficient.

Thank you for the suggestion. We opt to remain with the confidence intervals shown both as labels and error bars unless the Editor recommends otherwise.

7)     The results are clearly presented and discussed with validation.

Thank you for the generous comment.

Reviewer 3 Report

The manuscript entitled "Stiffness Evaluation of Laboratory and Plat Produced Foamed Bitumen Warm Asphalt Mixtures with Fiber Reinforcement and Bio-Flux Additive" demonstrates warm mix asphalt mixtures at 30C lower gained better performance on stiffness moduli than the mixtures at 15C lower based on experiments. The better performance is attributed to the inherent properties of foamed bitumen mixtures.

1. The introduction points out the "impact of water foaming" on the performance in minutes. Can the author also includes some part on the temperature influence? or is the experiment the first one to study temperature?

2. Can the authors elucidate why the viscoelastic responses changed depending on the loading times? Will the porosity change?

3. Based on Table 6, the p-values results of some factors are above 0.05, indicating no significant difference. Will it be helpful if remove the nonsignificant factors and do a re-fit?

4. Most of the citations are old more than five years. Can the authors include more recent studies on this topic?

Author Response

Dear Reviewer,

We would like to voice our gratitude for the time and effort spent revising our paper titled “Stiffness Evaluation of Laboratory and Plant Produced Foamed Bitumen Warm Asphalt Mixtures with Fiber Reinforcement and Bio-Flux Additive”. We truly feel that your remarks are a significant contribution to the overall quality of our paper and enabled us to rectify its shortcomings.

After a thorough revision, we present you the corrected version of the manuscript for its assessment. Please find the detailed responses to your comments below.

Best regards,

Authors

Reviewer 3

The manuscript entitled "Stiffness Evaluation of Laboratory and Plat Produced Foamed Bitumen Warm Asphalt Mixtures with Fiber Reinforcement and Bio-Flux Additive" demonstrates warm mix asphalt mixtures at 30C lower gained better performance on stiffness moduli than the mixtures at 15C lower based on experiments. The better performance is attributed to the inherent properties of foamed bitumen mixtures.

  1. The introduction points out the "impact of water foaming" on the performance in minutes. Can the author also includes some part on the temperature influence? or is the experiment the first one to study temperature?

Thank you for the comment. We have elaborated on the effects of water foaming and expanded L55-61:

“In previous studies it has been shown that the impact of water foaming on the performance properties and ageing characteristics of asphalt binders is small in technical terms and statistically non-significant in many of the investigated cases. In particular, no evidence for significant oxidative stress associated with asphalt binder foaming was found [18], and while the effects of foaming on viscoelastic properties of asphalt binders may be significant [19], they subside after short-term ageing [20]. Therefore, the effects of foaming on asphalt binders may be omitted in the mixture design, therefore simplifying it and not interfering with other possible additives.”

We have also elaborated on the effects of lowered processing temperatures of WMA mixtures in L72-78:.

  1. Can the authors elucidate why the viscoelastic responses changed depending on the loading times? Will the porosity change?

Thank you for the comment. We have attempted to clarify this matter in L157-164. The samples were tested at low strains (50-100 με) to to ensure linear viscoelastic response. It could be therefore assumed that the VE responses of the mixtures depended on loading times and testing temperature. Because the testing was conducted in compressive mode it was done from low to high temperatures and from high to low loading frequencies to mitigate the effects of permanent deformations and the changes in void contents (to which efforts the low strains also contributed constructively).

  1. Based on Table 6, the p-values results of some factors are above 0.05, indicating no significant difference. Will it be helpful if remove the nonsignificant factors and do a re-fit?

Thank you for the comment. In fitting of statistical models, it is good practice to reduce the models as much as it is possible by omitting the factors in the experiment which do not significantly contribute to the variance in the measured response. In our case the sole purpose of the statistical analysis was to evaluate which the factors in the experiment significantly influenced the complex stiffness moduli, therefore a reevaluation of the models was not needed.

  1. Most of the citations are old more than five years. Can the authors include more recent studies on this topic?

Thank you for the accurate comment. We have attempted to add more recent literature to the review.

Round 2

Reviewer 1 Report

Accept